# Non-HLA Antibodies in Kidney Transplantation: Immunity and Genetic Insights

**DOI:** 10.3390/biomedicines10071506

**Published:** 2022-06-25

**Authors:** Bogdan Marian Sorohan, Cătălin Baston, Dorina Tacu, Cristina Bucșa, Corina Țincu, Paula Vizireanu, Ioanel Sinescu, Ileana Constantinescu

**Affiliations:** 1Department of General Medicine, Carol Davila University of Medicine and Pharmacy, 020021 Bucharest, Romania; drcbaston@gmail.com (C.B.); umfisinescu@gmail.com (I.S.); ileana.constantinescu@imunogenetica.ro (I.C.); 2Center for Uronephrology and Kidney Transplantation, Fundeni Clinical Institute, 022328 Bucharest, Romania; ticu_dorina@yahoo.com (D.T.); cristinabucsa@yahoo.com (C.B.); corina_tarangoi@yahoo.fr (C.Ț.); luca_paula2000@yahoo.com (P.V.); 3Department of Immunogenetics, Fundeni Clinical Institute, 022328 Bucharest, Romania

**Keywords:** non-HLA, antibodies, kidney transplant, immunity, genetic, antigen, mismatch, AT1R, ETAR

## Abstract

The polymorphic human leukocyte antigen (HLA) system has been considered the main target for alloimmunity, but the non-HLA antibodies and autoimmunity have gained importance in kidney transplantation (KT). Apart from the endothelial injury, secondary self-antigen exposure and the presence of polymorphic alloantigens, respectively, auto- and allo- non-HLA antibodies shared common steps in their development, such as: antigen recognition via indirect pathway by recipient antigen presenting cells, autoreactive T cell activation, autoreactive B cell activation, T helper 17 cell differentiation, loss of self-tolerance and epitope spreading phenomena. Both alloimmunity and autoimmunity play a synergic role in the formation of non-HLA antibodies, and the emergence of transcriptomics and genome-wide evaluation techniques has led to important progress in understanding the mechanistic features. Among them, non-HLA mismatches between donors and recipients provide valuable information regarding the role of genetics in non-HLA antibody immunity and development.

## 1. Introduction

Kidney transplantation (KT) remains the elective method of renal replacement therapy in patients with end-stage renal disease [1]. The main factor limiting the long-term success of KT is the immunological barrier [2]. The polymorphic human leukocyte antigen (HLA) system has been considered the main target for alloimmunity. Thus, after several years of research, HLA typing, anti-HLA antibody detection techniques, donor-recipient matching and selective immunosuppression therapy have significantly improved kidney graft outcomes [3]. Anti-HLA donor specific antibodies are the main factor involved in the pathophysiology of antibody-mediated rejection (ABMR) [4]. However, in the last two decades, since the reported cases of rejection in the absence of anti-HLA antibodies, the importance of antibodies directed against antigens outside the major histocompatibility complex has begun to be recognized. Thus, the presence of non-HLA antibodies was identified, and autoimmunity began to be better understood in the pathology of rejection [5,6,7]. Several non-HLA antibodies were described against endothelial, epithelial or various proteins in association with negative kidney graft outcomes. In KT, the bulk of evidence for non-HLA antibodies comes from observational studies, in which antibodies against the following antigens have been described: angiotensin II type 1 receptor antibodies, endothelin-1 type A receptor, collagen type IV, fibronectin, perlecan, vimentin, agrin, major histocompatibility complex class I chain-related gene A and B, Rho guanine nucleotide exchange factor 2, peroxisomal trans-2-enoyl-CoA reductase, protein kinase C zeta type and H-Y antigen (Table 1) [8,9]. The focus of the KT community was centered on angiotensin II type 1 receptor antibodies (AT1R-Ab) [10,11]. Now, we know that alloimmunity and autoimmunity play an integrated and complementary role in the development of non-HLA antibodies, and the emergence of transcriptomics and genome-wide evaluation techniques has led to important progress in understanding the mechanisms of antibody formation [12,13]. The purpose of this review is to provide an evidence-based update regarding immune and genetic insights in the pathophysiology of non-HLA antibody formation, types of non-HLA antibodies and the role of non-HLA mismatch in KT.

## 2. Immunity Aspect of Non-HLA Antibodies

Non-HLA antibodies can be classified as alloantibodies (directed against polymorphic antigens that differ between the recipient and donor) or, more frequently, as autoantibodies (which recognize self-antigens that are usually cryptic) [6]. These antibodies recognize non-HLA antigens located in various tissues and cells such as endothelial vascular cells, smooth muscle vascular cells, tubular epithelial cells, podocytes, mesangial cells and immune cells [6,10,14]. As the vascular endothelium is the main component interposed between recipient immune system and the transplanted kidney, most of the antigens are endothelial autoantigens that trigger non-HLA autoantibody formation [12,14].

The mechanisms of non-HLA antibody formation are not fully understood. Both auto- and alloantibodies directed against non-HLA antigens share some common steps in the pathophysiological mechanism of their development. The proposed models are mainly based on basic research studies performed on animal or human cell cultures [6,13].

Non-HLA autoantibody development involves a first step of endothelial injury and exposure of neoantigens or cryptic antigens [12]. Before transplantation, this injury could be produced by one or more of the following factors: acute kidney injury, chronic kidney disease, lupus nephritis, focal segmental glomerulosclerosis, diabetes or preeclampsia [15,16,17,18,19,20,21,22]. In the post-transplant setting, development of non-HLA antibody formation is more complex and involves specific transplant-associated factors: different causes of endothelial injury (ischemia-reperfusion injury, BK nephropathy, acute graft pyelonephritis, anti-HLA antibodies, rejection, type of immunosuppression), appropriate conditions for autoimmunity through loss of self-tolerance and the interplay between autoimmunity and alloimmunity [6,12,23,24]. Following the endothelial injury process, along with the expression of cryptic autoantigens, alloantigens, DAMPs and extracellular vesicles are also exposed on the cell surface. Although all can participate in the process of autoantibody formation, autoantigens and extracellular vesicles are the most important because they activate antigenic recognition through the indirect pathway. Stimulation of the immune response through the indirect recognition pathway involves the recognition and processing of the antigen by the antigen-presenting cells of the recipient and their presentation to the T cells of the recipient. Activation of the indirect pathway leads to the selection of T cell autoreactive cells, predominantly a subpopulation of helper 17 cells and subsequently to the stimulation of autoreactive B cells responsible for antibody production. Key processes in autoantibody formation include the phenomenon of epitope spreading, loss of self-tolerance and cross-reactivity. Additionally, the interplay between alloimmunity and autoimmunity contributes decisively to the process described above. Extracellular vesicles are cell-derived membrane structures with antigenic content, including autoantigens, which are capable of stimulating immune recognition via indirect pathways. T helper 17 cells are a subset of T lymphocytes involved in the formation of chronic inflammation and the pathogenesis of autoimmune diseases. Their predominant activation influences inflammatory cell recruitment, tertiary lymphoid organ formation, B cell differentiation, B cell maturation and altered self-tolerance. These events are implicated in the occurrence of inflammation and autoantibody formation (Figure 1) [6,10,12,14,23,25,26].

Non-HLA alloantibody development does not necessarily require endothelial injury. This process involves a mismatch of non-HLA antigens between the donor and recipient, which is sufficiently antigenic to induce an immune response against the non-self-antigen following transplantation. In this case, allorecognition is produced by recipient antigen-presenting cells through an indirect pathway of recognition, and the following steps are similar to those for autoantibody formation. A classic example of non-HLA alloantibody formation could be found in gender mismatch KT (H-Y antigen) when female recipients receive a graft from male donors (Figure 1) [6,13,27].

Recently, it has been shown that innate-like B cells (Bin cells) could act as local drivers of non-HLA immunity. Asano and colleagues showed that infiltrating Bin cells of the kidney graft had a unique signature represented by AHNAK and AHNAK covariant genes, expression of IL-15 and type-1 IFN. Intrarenal Bin cells expressed highly mutated IgG autoantibodies, which do not bind donor-specific antigens, nor are they enriched for reactivity to ubiquitously expressed self-antigens, but are reactive with either renal specific or inflammation-associated antigens. Furthermore, local antigens can drive Bin cell proliferation and differentiation into plasma cells expressing self-reactive antibodies. Moreover, they showed that this type of B cell is involved in the loss of peripheral tolerance to organ-restricted antigens. Based on their inflammatory state, and antibody specificity, Bin cells connect innate and adaptive immunity to drive local inflammation [28,29].

## 3. Types of Non-HLA Ab in Kidney Transplantation

### 3.1. Angiotensin II Type 1 Receptor Antibodies (AT1R-Ab)

AT1R-Ab are the most widely known and researched type of non-HLA antibody, which generated the most interest in the KT community in the last 15 years [6,10]. The presence of AT1R-Ab was described for the first time by Dragun et al. in 2005 [30]. In that seminal paper, researchers observed that AT1R-Ab were associated with malignant hypertension, steroid-refractory vascular rejection and decreased graft function [30]. Following this discovery, an important number of studies has analyzed different outcomes among kidney transplant recipients with AT1R-Ab [10,31]. The presence of these antibodies, both pre-transplant and post-transplant, has been shown to be associated with the development of different rejection phenotypes, as well as with a negative impact on renal graft function and survival [10].

The prevalence of AT1R-Ab in KT ranges from 2.1% to 59%. This large variability could be influenced by study design, immunological risk of the KT recipients, immunosuppression type, time of antibody assessment and positivity threshold for detection [10].

AT1R-Ab are considered autoantibodies and belong to the IgG1 and IgG3 complement fixing subclass [30,32]. The mechanism of AT1R-Ab formation entails a chronological sequence of pathophysiological events, superimposed over those involved in autoantibody formation [10].

AT1R-Ab recognize two different epitopes, AFHYESQ and ENTN, from the second extracellular loop of the AT1R, a transmembrane protein belonging to the family A G-protein-coupled receptors, highly expressed by the endothelial cells and smooth muscle cells of the renal vasculature, mesangial cells, podocytes, proximal tubular epithelial cells and interstitial cells from the inner strip of the outer medulla (Figure 2) [23,30,33,34,35]. Intrarenal expression of AT1R is variable and influenced by genetic and non-genetic factors. Among the genetic factors, A1166C polymorphism of the AGTR1 gene is a well-known determinant of increased tissue levels of the receptor [36,37]. AT1R-Ab determines an agonistic, allosteric effect on AT1R in a sustained manner, which leads to pathological hyperactivation of the receptor, responsible for the appearance of the inflammatory and procoagulant environment via activation of intracellular pathways. Despite being IgG1 and IgG3 antibodies, AT1R-Ab seem to produce endothelial injury by a complement-independent mechanism, through direct endothelial cell activation and recruitment of inflammatory cells to the graft [23,30]. The activated G protein-dependent signaling pathways are the phospholipase C pathway and the PKC pathway. Upon activation, they induce a cellular response based on vasoconstriction of the smooth muscle cells and proliferation of vascular endothelial and smooth muscle cells. As for G protein-independent signaling pathways, the key determinant in the development of rejection lesions is the phosphorylation of the Ras/Raf/MEK/ERK1/2 cascade. This exerts its role through certain transcription factors such as AP-1 and NFkB, responsible for increasing gene expression of pro-inflammatory (MCP-1 and RANTES) and procoagulant factors (tissue factor) [30,38,39]. MCP-1 and RANTES are important mediators of the proinflammatory immune response, being involved in the formation of a series of cytokines (IL-1, IL-6, IL-8, IL-17, TNF-α, IFN-γ) that participate in the formation of microvascular inflammation in the glomerulus and peritubular capillaries [30,38,40]. Stimulation of endothelial growth factor release activates the coagulation cascade and promotes thrombus formation [30,41]. The stimulation of AT1R present on the surface of inflammatory cells, especially polymorphonuclear, monocytes and B and T lymphocytes amplifies the microvascular inflammatory response, which may result in endothelial injury and additional AT1R expression, thus forming a vicious circle responsible for graft injury [10,42,43,44]. Vasoconstriction, vascular endothelial and smooth muscle cell proliferation, microvascular inflammation and thrombosis can lead to endarteritis, fibrinoid necrosis of the vascular wall, subintimal fibrosis and vascular occlusion, which are elements associated with vascular rejection [10,30,32,45,46].

Patients with AT1R-Ab have an increased risk of developing different rejection phenotypes (T-cell mediated, antibody-mediated or vascular rejection) and also have decreased short-term and long-term graft function and survival. The impact of antibodies on graft function may be explained indirectly through rejection or directly through stimulation of inflammation and fibrosis. A persistent relationship has been demonstrated between the presence of AT1R-Ab and a vascular proinflammatory cytokine profile (TNF-α, IFN-γ, IL-8, IL-1β, IL-6 and IL-17) with negative implications on graft function, even in the absence of rejection [40]. In a recent study, Betjes et al. demonstrated that AT1R-Ab have been associated with the development and progression of interstitial fibrosis and graft loss, independent of the presence of rejection [47]. Special consideration should be given to AT1R-Ab and HLA-DSA double positivity, observed more frequently in highly sensitized patients. In this scenario, the antibodies have a synergistic effect with a negative impact on graft function and survival [12,16,23].

### 3.2. Anti-Endothelin a Receptor Antibodies

Endothelin A receptor (ETAR) is a transmembrane G-protein-coupled receptor, encoded by the ENDRA gene, located on chromosome 4, which is involved in blood pressure and sodium balance homeostasis [48]. ETAR as a non-HLA antigen has led to interest in KT with the identification of other anti-endothelial cell antibodies. ETAR is mainly expressed on vascular endothelial cells, vascular smooth muscle cells, mesangial cells and tubular epithelial cells, but also on immune cells [44,49]. Their expression could be influenced by genetic or environmental factors. Regarding genetic factors, polymorphic variants of the gene have been described in association with an increased ETAR expression [50,51].

Anti-ETAR antibodies are IgG1 subtype autoantibodies which bind EQHKTCMLNATSK sequence from the second extracellular loop of ETAR and act in an agonist, persistent manner, comparable to AT1R-Ab [50]. The prevalence of anti-ETAR antibodies in KT recipients was reported in up to 47.4% [52]. The mechanisms of antibody formation are similar to those of AT1R-Ab, where endothelial injury and autoimmunity are necessary conditions [10,50].

Some studies have proven that anti-ETAR antibodies are associated with ABMR, vascular rejection, worse graft function in the first year after KT and graft loss [52,53,54]. Moreover, one study has reported an association between anti-ETAR antibodies and AT1R-Ab and a worse graft function for those with double positivity [54].

The phenotype of anti-ETAR antibody mediated graft injury has not been clearly established yet. Two mechanisms through which anti-ETAR antibodies could cause graft injury have been proposed: complement-dependent and independent [55]. The complement-dependent mechanism involves activation of the classical complement pathway by IgG1-type anti-ETAR antibodies, similar to anti-HLA antibodies, and graft biopsies in these patients show linear C4d fixation in peritubular capillaries on immunofluorescence examination. The complement-independent mechanism is similar to AT1R-Ab, which involves endothelial cell activation and recruitment of inflammatory cells to the graft through ERK signaling pathways, and graft biopsy showed an absence of C4d in immunofluorescence in this scenario. Considering the similarities involved in the vascular lesions, such as endarteritis, fibrinoid necrosis and vascular occlusion secondary to inflammation, vasoconstriction, vascular endothelial and smooth muscle cell proliferation, the histological pattern is probably similar to that determined by AT1R-Ab [10,23,55]. Recently, in a basic research study, Catar et al. showed that anti-ETAR and AT1R-Ab antibodies could impact microvascular endothelium via β2-arrestin and mTORC2 pathway activation, in a PIK3-dependent and ERK-independent manner [56].

### 3.3. Anti-Mica Antibodies

The major histocompatibility complex (MHC) class I-related chain A (MICA) antigen is a highly polymorphic glycoprotein encoded by MHC class I-related chain, a gene located on chromosome 6, within the HLA complex, centromeric to HLA-B locus at 46.4 kb. The MICA molecule is expressed on endothelial, epithelial, monocyte and dendritic cells, but not on T or B lymphocytes [57]. Moreover, MICA could bind the ligand for activating the C-type lectin-like receptor (NKG2D) expressed on natural killer (NK) cells, γδ T-lymphocytes, and CD8+ αβ T-lymphocytes, producing their activation [58].

Antibodies directed against the MICA glycoprotein were among the first non-HLA antibodies reported in KT. Among the first and most important studies to evaluate the role of anti-MICA antibodies in KT was that of Zou et al. [59]. Subsequently, several studies have examined the importance of anti-MICA antibodies in KT and have shown that pre-transplant or post-transplant antibodies can be associated with the development of ABMR or TCMR and that they negatively impact kidney graft function [60,61,62,63,64]. The prevalence of anti-MICA antibodies in KT can be as high as 20% [65]. Preformed anti-MICA antibodies could be formed by classic sensitizing events such as pregnancy, blood transfusions and previous transplants [66]. Additionally, anti-MICA antibodies could be found in the absence of sensitizing events that make the development of these antibodies through autoimmunity phenomena likely [66]. A mismatch between donor and recipient MICA alleles could lead to de novo anti-MICA donor-specific antibody formation, independent of HLA immunity [63]. These antibodies seem to exhibit synergistic action with anti-HLA antibodies, increasing the risk of rejection and graft loss [63]. The possible mechanism of anti-MICA antibody graft-mediated injury could be produced through NKG2D-mediated cytotoxicity and complement-mediated cytotoxicity [57].

### 3.4. Anti-Perlecan/LG-3 Antibodies

Perlecan is a heparan sulfate proteoglycan present in vascular and epithelial basement membranes. The C-terminal fragment of perlecan contains LG3 (laminin-like globular domain), which is considered highly immunogenic [67]. Increased levels of serum cathepsin L, LG3 and urinary LG3 were reported in KT recipients with vascular rejection, which suggested LG3 as both a biomarker and a pathophysiological element of vascular injury [68,69]. In the presence of endothelial dysfunction and apoptosis, caspase 3 activates cathepsin L, which cleaves LG3 domain and releases LG3 fragments as apoptotic exosome-like vesicles (ApoExo). These ApoExe containing LG3 are immunogenic and fuel anti-LG3 antibody production [67].

The appearance of anti-LG3 antibodies follows the principles of autoimmunity and may be determined by endothelial injury due to uremia, ischemia-reperfusion injury, treatment with calcineurin inhibitors or rejection [67]. Cardinal et al. showed that anti-LG3 antibodies were IgG1 and IgG3 fixing and activating antibodies [70]. It has been shown that anti-LG3 antibodies were aggravating factors for ischemia-reperfusion injury, and these antibodies were associated with immune-mediated vascular rejection, delayed-graft function and lower graft function at 1 year after transplantation [70,71]. The potential mechanisms of graft injury driven by anti-LG3 antibodies seem to be complement-dependent and are based on the following basic research findings: C4d deposition, microvascular inflammation in the peritubular capillaries, neointima formation subsequent to vascular smooth muscle cells of the recipient-derived mesenchymal stem cells, obliterative vascular remodeling and kidney graft fibrosis [68,70,71,72]. Recent insights into immune mechanisms of antiperlecan/LG3 antibody production showed that memory B cells against LG3 are T cell independent, but that production of anti-LG3 antibodies requires the help of T cells. The important role of T lymphocytes in anti-LG3 antibody production is also supported by the decrease in antibody titers after initiation of immunosuppressive treatment with calcineurin inhibitors [73].

### 3.5. Anti-Agrin Antibodies

Agrin is a heparan sulfate proteoglycan highly expressed in glomerular basement membranes [74]. C-terminal fragment (CAF), a cleavage product of agrin, was evaluated as a potential biomarker for graft function in KT recipients. It has been shown that CAF was correlated with creatinine, cystatin C, estimated glomerular filtration rate, early after KT and also with delayed graft function [75,76]. In another study, post-transplant CAF was identified as a risk factor for worsening proteinuria and kidney graft loss [77]. Antibodies against agrin were described in patients with transplant glomerulopathy, a feature of chronic ABMR [78]. Anti-agrin antibodies could be classified as autoantibodies, which are formed after agrin expression as a neoantigen.

### 3.6. Anti-Collagen Type Iv, Type Iii, Type I and Anti-Fibronectin Antibodies

Collagen type IV and fibronectin are important constituents of glomerular basement membrane, and in circumstances of tissue injury they became self-antigens. Preformed and de novo antibodies against collagen type IV and fibronectin increase the risk of transplant glomerulopathy development. Patients with TG, anti-collagen type IV and anti-fibronectin antibodies had a significant activation of collagen-IV and fibronectin specific T cells secreting IFN-γ and IL-17 and a significant decrease in specific IL-10 secreting T cells. The aforementioned events explain the immune response against self-antigens and loss of peripheral tolerance [79]. Moreover, in a recent study, Sehoon Park and colleagues showed that anti-collagen type I and type III antibodies were significantly higher and were associated with an increased risk of death-censored graft failure in ABMR. Based on these results, the authors proposed type III and type I anti-collagen antibodies as biomarkers with a diagnostic and prognostic role in ABMR [80].

### 3.7. Anti-Vimentin Antibodies

Vimentin is a cytoskeletal type 3 intermediate filament protein expressed in endothelial cells, smooth muscle cells, epithelial cells, T cells, neutrophils, fibroblasts and platelets [81]. Usually, vimentin is an intracellular molecule, but, in the context of endothelial injury or apoptosis, it is expressed on the cell surface and becomes an immunogenic autoantigen, which stimulates anti-vimentin antibody formation [82].

Patients with graft failure have an increased presence of anti-vimentin antibodies and the titer of antibodies is higher in those with anti-HLA DQ2 donor-specific antibodies [83]. Anti-vimentin antibodies were reported in relation to two major histopathological lesions: IFTA (interstitial fibrosis and tubular atrophy) and chronic ABMR [84,85]. These antibodies seem to bind and activate complement given the presence of C4d in peritubular capillaries in those with chronic ABMR [86].

### 3.8. Anti-H-Y Antibodies

The H-Y antigen is a minor histocompatibility antigen encoded by genes located on chromosome Y. The genes for H-Y antigens encode 1 to 245 polymorphisms of the proteins RPS4Y1, DDX3Y, UTY, SMCY, which highly promotes immunogenicity and specificity [87]. It has been shown that in gender mismatched KT, female recipients who received grafts from male donors developed alloantibodies against RPS4Y1 and DDX3Y H-Y antigens [87]. Furthermore, the presence of these alloantibodies was reported in association with graft rejection and short- and long-term graft failure [27,87,88].

### 3.9. Anti-ARHGDIB (Rho Guanine Nucleotide Exchange Factor 2) Antibodies

ARHGDIB is an intracellular GTP-binding protein, which is involved in many cellular activities and widely expressed in different tissues and organs. Expression in the kidney graft is influenced by normal or pathological states. In a kidney graft without histological abnormalities, ARHGDIB expression is weak in endothelial cells of interlobular arteries, peritubular capillaries and glomerular capillaries. Contrarily, in a kidney graft with acute tubular necrosis, ARHGDIB expression is robust in endothelial cells of interlobular arteries, peritubular capillaries and glomerular capillaries. In this scenario, ARHGDIB is also seen in podocytes and lymphocytes [89].

In a nationwide cohort study of 4770 KT recipients, pre-transplant anti-ARHGDIB antibodies were found to be a significant risk factor for graft loss in those who received a graft from a deceased donor, independent of anti-HLA DSA. This observation suggests the role of endothelial injury, created by ischemia-reperfusion injury, in increasing extracellular expression of ARHGDIB and autoantibody formation [89]. In another study, anti- ARHGDIB antibodies were significantly associated with decreased graft survival, to which patients with concomitant anti-HLA DSA were more prone [9]. The same study showed that anti-ARHGDIB antibodies were not associated with increased rejection rates of any histological phenotype, but the ARHGDIB gene was overexpressed in patients with biopsy-proven ABMR [9]. In an opposite manner, Betjes and colleagues found that anti-ARHGDIB autoantibodies were significantly increased in patients with chronic active ABMR, but were not associated with graft survival [47].

### 3.10. Anti-PECR (Peroxisomal Trans-2-Enoyl-CoA Reductase) Antibodies

PECR is a peroxisomal NADPH-specific trans-2-enoyl-CoA reductase that catalyzes the reduction of trans-2-enoyl-CoAs of varying chain lengths from 6:1 to 16:1, having maximum activity with 10:1 CoA [90]. This peroxisomal protein plays an important role in fatty-acid biosynthesis [91]. Its expression is increased in the kidney, endothelial cells and immune cells [92]. Different injuries of the graft could lead to surface exposure of this intracellular protein [92]. In KT, PECR was described in association with the development of transplant glomerulopathy and biopsy-proven ABMR [9,92].

### 3.11. Anti-PRKCZ (Protein Kinase C Zeta Type) Antibodies

PRKCZ is a type of protein-kinase C involved in processes of proliferation, apoptosis and cell survival and it also plays a regulatory role in inflammation [93]. In an animal model, the expression of PRKCZ was overexpressed after ischemia-reperfusion injury [94]. Sutherland et al. revealed that antibodies against PRKCZ were associated with graft rejection and loss. However, the study conclusion was that there are not enough data to determine whether these antibodies were bystanders or pathogenic factors involved in graft rejection [93].

## 4. Non-HLA Mismatch

Non-HLA antigens are products of allograft-expressed donor genes that carry non-synonymous single nucleotide polymorphisms (nsSNPs) generating polymorphic peptides that are recognized as non-self by the immune system. Non-HLA antigen mismatch between donor and recipients (D-R) triggers the alloimmune system response, which is responsible for antibody formation. Moreover, specific variants in the non-HLA system and genetic ancestry of the donor or recipient are important contributors for the immune response and different kidney graft outcomes. The following non-HLA gene variant polymorphisms for the donor and recipient were reported in KT and had a negative impact on graft outcomes: donor gene variant polymorphism (APOL1, ABCB1, SHROOM3, CAV1 and UMOD) and recipient gene variant polymorphism (APOL1, LIMS1, FOXP3, TNFA and IFNG) [95].

Several studies which have evaluated non-HLA mismatches between D-R pairs, using genome-wide analysis, reported genetic insights in the understanding of non-HLA immunity. The study of Mesnard et al. was the first to analyze the importance of non-HLA D-R mismatch in a cohort of 53 D-R pairs, in KT from living donors, using whole-exome sequencing of DNA. They quantified the D-R mismatch based on the allogenomics mismatch score, by computing the number of amino acid mismatches in trans-membrane proteins. The allogenomics concept sums contributions of many mismatches that can impact protein sequence and structure and could engender an immune response in the graft recipient. Immunological and biophysical principles strongly suggest that alleles present in the donor genome, but not in the recipient genome, will have the potential to produce epitopes that the recipient immune system will recognize as non-self. The authors showed that the allogenomics mismatch score was a robust predictor of long-term graft function, independent of HLA-A, B, DR matching, donor age and time post-transplantation [96]. In another study, Pineda and colleagues tested the role of non-HLA D-R mismatches in kidney graft rejection, in a cohort of 28 pairs, using exome sequencing and gene expression data. They identified 123 non-HLA variants significantly associated with ABMR, TCMR and no rejection. Ninety-four out of 123 variants were associated with ABMR and 25 with TCMR. Eighteen genes responsible for multiple non-HLA variants were described as biologically relevant for rejection (AP3D1, CDC123, CDYL2, CSMD3, FAM129B, MUC3A, MYOM2, OR51F1, OR8G1, OR8G5, PNPLA6, PSEN2, RASA3, ZNF280D, AIM1L, CHRNA10 and KIAA1755 and SLC-family) and 15 among them were associated with the risk of post-transplant ABMR [97].

Roman Reindl-Schwaighofer et al. genotyped 477 KT recipients with stable graft function and their deceased donors, at three months after KT, and evaluated genome-wide mismatches in nsSNPs. This analysis was used to identify incompatibilities in transmembrane and secreted proteins. They analyzed 59,268 nsSNPs and found a median of 1892 (IQR 1850–1936) mismatches between D-R, affecting a transmembrane or secreted protein. The degree of nsSNP mismatch, adjusted for HLA eplet mismatch, was independently associated with graft loss. Furthermore, they used a customized peptide array, which contained self and non-self peptides, to test the donor-specific alloimmune response to genetically predict mismatched epitopes in a subset of 25 patients with chronic ABMR. They observed 16 non-HLA donor-specific antibody responses directed against the genetically predicted nsSNP mismatches in membrane-associated proteins [98]. In a cohort of 385 D-R pairs, Zhang et al. analyzed the role of D-R genetic differences in graft histology and survival, using genome-wide single nucleotide polymorphism (SNP) array data, excluding the HLA region. They used ADMIXTURE analysis to quantitatively estimate ancestry in each D-R pair and PLINK to estimate the proportion of genome-shared identity-by-descent (pIBD) between D-R pairs. In D-R pairs of similar ancestry, pIBD was significantly associated with allograft survival, independent of HLA mismatches. Moreover, pIBD was significantly inversely correlated with early (<1 year after KT) vascular intimal fibrosis, which was an independent predictor of graft survival [99].

## 5. Future Perspectives

At this moment, the immunological risk evaluation of KT recipients is based on the alloimmunity and anti-HLA antibody evaluation [100,101]. ABMR due to anti-HLA antibodies benefits from some evidence regarding evaluation and treatment, and new randomized clinical trials testing molecules such as interleukin-6 antagonism, CD38-targeting antibodies and selective complement inhibitors are in progress [100,102]. However, it is quite clear that the immunological risk assessment focused only on anti-HLA donor-specific antibodies is insufficient. From a clinical perspective, the incorporation of non-HLA antibodies into the immunological risk assessment could bring important benefits in the management of rejection cases. Although there are numerous studies, especially on anti-AT1R antibodies, that have demonstrated the role of non-HLA antibodies in rejection and the impact on kidney graft survival and function, there are still no recommendations and algorithms for risk assessment mediated by these antibodies in clinical practice [10,65]. Currently, the therapeutic approach of non-HLA antibody-mediated graft injury is based on the evidence from observational studies and case reports and consist of several solitary or combination regimens of plasma therapy, intravenous immunoglobulins, corticosteroids, antithymocyte globulin, rituximab, eculizumab, bortezomib and AT1R blockers [103]. A careful analysis of the previous studies, continuous development of genetic analysis, use of a molecular microscope diagnostic system, use of standardized assays and clinically relevant cutoff values, conducting randomized clinical trials and the use of therapeutic protocols may be the basis for the implementation of these antibodies in practice. All these advances could lead to the optimal management of non-HLA antibody-mediated graft injury by judicious use of the aforementioned drugs, but also by using promising molecules such as imlifidase, C1 esterase inhibitors, tyrosine kinase inhibitors or anti-CD38 antagonists.

## 6. Conclusions

The importance of non-HLA antibodies is gaining recognition in KT. Insight processes regarding the formation mechanisms and immune and genetic aspects of these antibodies are continuously developing and doubtlessly contribute to a better integrated understanding of alloimmunity and autoimmunity in graft injury. In practice, of paramount recent importance are the expansion of non-HLA antibody testing methods and the use of genome-wide techniques to evaluate non-HLA D-R mismatches. Thus, these advances can underpin both future research directions and the development of targeted therapeutic approaches in the field of non-HLA immunity.

## Figures and Tables

**Figure 1 biomedicines-10-01506-f001:**
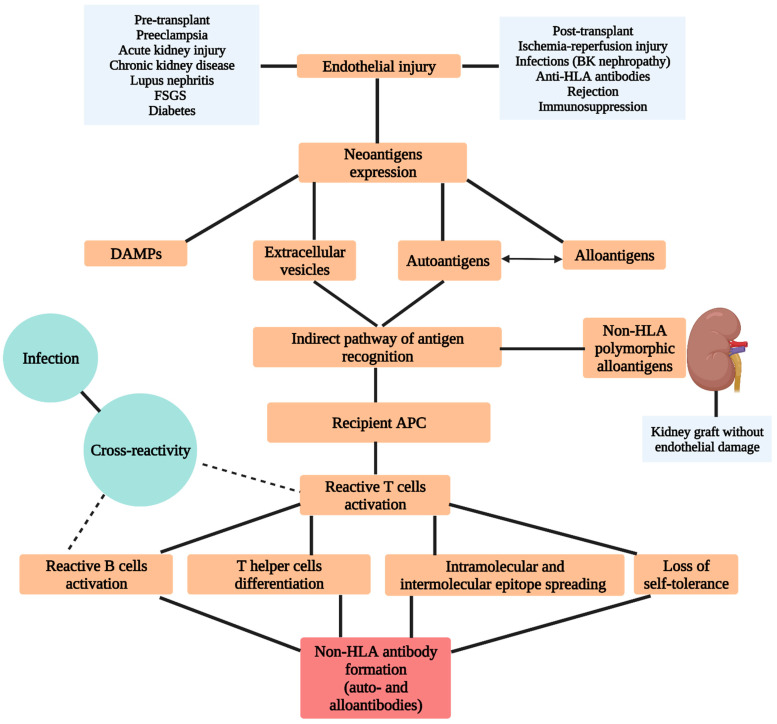
Mechanisms of non-HLA antibodies formation; FSGS–focal segmental glomerular sclerosis; BK–BK virus; HLA–human leukocyte antigen; DAMPs–damaging molecular patterns; APC–antigen presenting cell.

**Figure 2 biomedicines-10-01506-f002:**
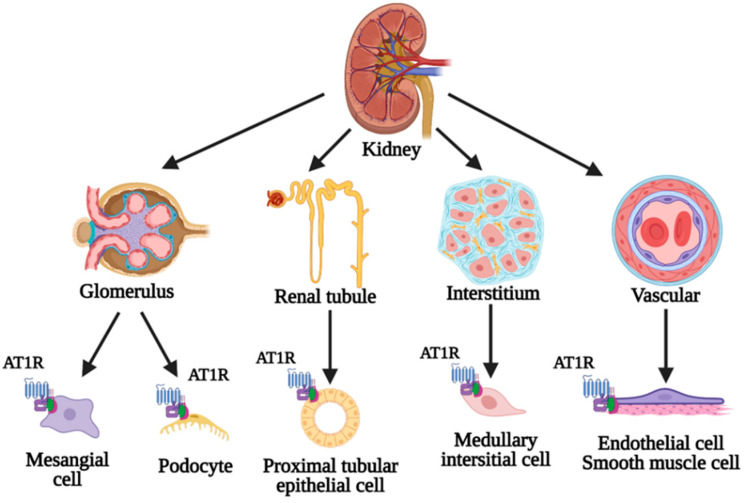
Intrarenal distribution of angiotensin II type I receptor (AT1R).

**Table 1 biomedicines-10-01506-t001:** Types of non-HLA antibodies.

Antibody Type	Autoantibody/Alloantibody	Mechanism of Injury
Anti-AT1R-Ab	Autoantibody	Complement independent
Anti-ETAR-Ab	Autoantibody	Complement independent
Anti-MICA-Ab	Autoantibody and Alloantibody	Complement dependent
Anti-perlecan-Ab	Autoantibody	Complement dependent
Anti-agrin-Ab	Autoantibody	Complement dependent
Anti-collagen type IV, III and I-Ab	Autoantibody	-
Anti-fibronectin-Ab	Autoantibody	-
Anti-vimentin-Ab	Autoantibody	Complement dependent
Anti-H-Y-Ab	Alloantibody	Complement dependent
Anti-ARHGDIB-Ab	Autoantibody	Complement dependent
Anti-PECR-Ab	Autoantibody	Complement dependent
Anti-PRKCZ-Ab	Autoantibody	-

Ab—antibody; AT1R—angiotensin II type 1 receptor; ETAR- endothelin A receptor; MICA—Major histocompatibility complex class I–related chain antigens; ARHGDIB- Rho guanine nucleotide exchange factor 2; PECR—Peroxisomal trans-2-enoyl-CoA reductase; PRKCZ—Protein kinase C zeta type.

## Data Availability

Not applicable.

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
