# Peer review of "Non-HLA Antibodies in Kidney Transplantation: Immunity and Genetic Insights"

_biomedicines, 2022, doi:10.3390/biomedicines10071506_

Round 1

Reviewer 1 Report

HLA matching of donor-recipient is a very important factor in allograft outcome. The greater the degree of HLA mismatching, the higher the risk of rejection, resulting in a poor prognosis for allograft outcome. In particular, HLA mismatching of donor-recipient causes the development of donor-specific anti-HLA antibody (DSA), which causes rejection. However, even in the absence of DSA, the form of antibody-mediated rejection can be seen in pathologic findings, so there is a lot of interest in non-HLA antibodies recently. Therefore, it is a meaningful paper. This review article is judged to be a well-written article based on various analysis.

Author Response

We are grateful for your feedback on the article.

Reviewer 2 Report

In general, the review article is complete. There are some minor issues that could be recovered better like the extracellular matrix antibodies like ECM1 were not observed. The last part of the text concerning HLA mismatch should be analyzed better at the end the text is confusing. The importance of pIBD should be amplified. The authors should discuss the relevance of anti-collagen antibodies doi 10.3389/ti.2022.10099

The proposed stratification of alloimmune risk should be discussed see doi 10.3389/ti.2022.10138 

The authors may include future perspectives on the field since there are several therapies which are being tested.  See doi 10.1080/14728214.2022.2091131

Author Response

We are grateful for the evaluation of the manuscript, but especially for the recommendations.
Regarding the extracellular matrix request, according to our knowledge and after a literature search there are no published articles about ECM1 antibodies in kidney transplantation.
We modified some sentences in the non-HLA mismatch section,
for more clarity and for a better understanding of the text 
We added the relevance of anti-collagen type I and III antibodies, from the recommended article ( Section 3. Non-HLA antibodies in kidney transplantation, subsection, 3.6. Anti- collagen type IV, type III, type I and anti- fibronectin antibodies).
According to your recommnendations we decided to add a new section (5. Future perspective) were we discuss the stratification of the immunological risk and about several therapies in ABMR, using the recommended articles.